# LLM2Vec: Large Language Models Are Secretly Powerful Text Encoders

**Parishad BehnamGhader**[*,◇]   **Vaibhav Adlakha**[*,◇,†]   **Marius Mosbach**[◇]
**Dzmitry Bahdanau**[†]   **Nicolas Chapados**[†]   **Siva Reddy**[◇,†,‡]

[◇]McGill University, Mila   [†]ServiceNow Research   [‡]Facebook CIFAR AI Chair

{parishad.behnamghader,vaibhav.adlakha,marius.mosbach}@mila.quebec

## Abstract

Large decoder-only language models (LLMs) are the state-of-the-art models on most of today's NLP tasks and benchmarks. Yet, the community is only slowly adopting these models for text embedding tasks, which require rich contextualized representations. In this work, we introduce LLM2Vec, a simple unsupervised approach that can transform any decoder-only LLM into a strong text encoder. LLM2Vec consists of three simple steps: 1) enabling bidirectional attention, 2) masked next token prediction, and 3) unsupervised contrastive learning. We demonstrate the effectiveness of LLM2Vec by applying it to 4 popular LLMs ranging from 1.3B to 8B parameters and evaluate the transformed models on English word- and sequence-level tasks. We outperform encoder-only models by a large margin on word-level tasks and reach a new unsupervised state-of-the-art performance on the Massive Text Embeddings Benchmark (MTEB). Moreover, when combining LLM2Vec with supervised contrastive learning, we achieve state-of-the-art performance on MTEB among models that train only on publicly available data (as of May 24, 2024). Our strong empirical results and extensive analysis demonstrate that LLMs can be effectively transformed into universal text encoders in a parameter-efficient manner without the need for expensive adaptation or synthetic GPT-4 generated data.

## 1 Introduction

Text embedding models aim to encode the semantic content of natural language text in vector representations which then facilitate various natural language processing (NLP) tasks, such as semantic textual similarity, information retrieval, and clustering. For many years, the dominating paradigm for building such models relied on pre-trained bidirectional encoders or encoder-decoders such as BERT (Devlin et al., 2019) and T5 (Raffel et al., 2020), which are typically adapted for text embedding tasks by following a multi-step training pipeline consisting of weakly- and fully-supervised contrastive training (Ni et al., 2022; Li et al., 2023a; Xiao et al., 2023, *inter alia*). Only recently, the community started to adopt decoder-only LLMs for embedding text (Muennighoff, 2022; Ma et al., 2023; Wang et al., 2023; Springer et al., 2024; Li & Li, 2024).

We speculate that the slow adoption of decoder-only LLMs for text embedding tasks is partly due to their causal attention mechanism, which inherently limits their ability to produce rich contextualized representations. At any given layer, causal attention limits token interactions, ensuring that the representation of a token at position $i$ is influenced solely by the representations of preceding tokens at positions $0, 1, \ldots, i-1$. Although this limitation is necessary for generative capabilities, it is sub-optimal for text embeddings as it prevents the representations from capturing information across the entire input sequence.

Overcoming this architectural limitation of decoder-only LLMs for text embedding tasks is highly appealing as these models come with several advantages compared to their encoder-

---

[*]Equal contribution.

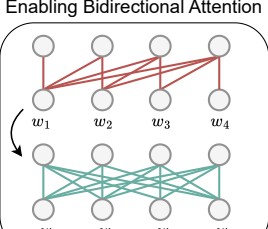 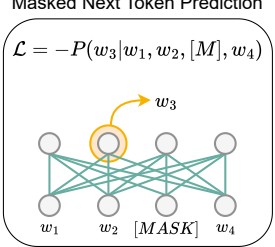 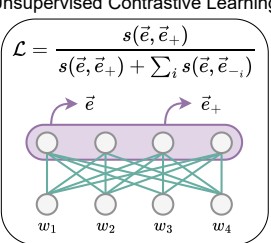

Figure 1: The 3 steps of LLM2Vec. First, we enable bidirectional attention to overcome the restrictions of causal attention **(Bi)**. Second, we adapt the model to use bidirectional attention by masked next token prediction training **(MNTP)**. Third, we apply unsupervised contrastive learning with mean pooling to learn better sequence representations **(SimCSE)**.

only counterparts.[1] During pre-training, decoder-only LLMs learn from all input tokens and not just a small percentage[2], which—given the same amount of training data—makes them much more sample-efficient than encoder-only models (Clark et al., 2020). Moreover, there exists a rich ecosystem around these models, with extensive tooling and well tested pre-training recipes, which has resulted in continuous improvement of these models by the community. Lastly, recent work on instruction fine-tuning and learning from human preferences has resulted in decoder-only LLMs that excel at instruction following (Wang et al., 2022b; Ouyang et al., 2022), making them an ideal choice for building *universal text embedding models* that generalize across a large variety of tasks using instructions.

In this work, we provide a simple unsupervised approach, termed **LLM2Vec**, which can be used to transform *any* pre-trained decoder-only LLM into a (universal) text encoder. As shown in Figure 1, LLM2Vec consists of three simple steps: 1) enabling bidirectional attention, 2) masked next token prediction, and 3) unsupervised contrastive learning. Crucially, LLM2Vec does not require any labeled data and is highly data- and parameter-efficient.

We apply LLM2vec to 4 decoder-only LLMs ranging from 1.3B to 8B parameters (`S-LLaMA-1.3B`, `LLaMA-2-7B`, `Mistral-7B`, `Meta-LLaMA-3-8B`) and evaluate the resulting models on word- and sequence-level tasks. On word-level tasks (chunking, named-entity recognition, and part-of-speech tagging), LLM2Vec-transformed models outperform strong encoder-only models by a large margin, demonstrating its effectiveness for producing rich contextualized token representations. On the Massive Text Embeddings Benchmark (MTEB), LLM2Vec-transformed models set a new state-of-the-art for unsupervised models, with our best model reaching a score of 56.8. Additionally, we combine LLM2Vec with supervised contrastive training and achieve a new state-of-the-art performance among models that train only on publicly available data. Beyond our strong empirical results, we provide an extensive analysis of how LLM2Vec affects the representations of the underlying model and reveal an intriguing property of Mistral-7B, which can handle bidirectional attention without any fine-tuning.

Overall, our work demonstrates that decoder-only LLMs are indeed capable of producing universal text embedding and only very little adaptation is required to reveal this ability. Our code and pre-trained models is publicly available at `https://github.com/McGill-NLP/llm2vec`

---

[1]We acknowledge that there are also several challenges associated with the large size of these models and provide a discussion in Appendix A.

[2]Encoder-only models are typically pre-trained by masking 15% of the tokens in the input sequence (Devlin et al., 2019).

## 2 LLM2Vec

### 2.1 Three simple ingredients

**Enabling bidirectional attention** The first step of the LLM2Vec approach is to replace the causal attention mask of decoder-only LLMs by an all-ones matrix (see Appendix B.1 for background on the self-attention). This gives each token access to every other token in the sequence, converting it into a bidirectional LLM (Devlin et al., 2019; Liu et al., 2019). However, it is not a priori clear, why this should lead to better sequence representations. After all, the decoder-only LLM was not trained to attend to future tokens and therefore, this naive approach might even lead to worse representations. As we show, simply enabling bidirectional attention does indeed decrease in embedding performance for most models. We can however easily adapt a model to make use of its bidirectional attention.

**Masked next token prediction** We use a simple strategy to make the model aware of its bidirectional attention by adapting it via *masked next token prediction* (MNTP). MNTP is a training objective that combines next token prediction with masked language modeling (Lv et al., 2023). Given an arbitrary sequence $\mathbf{x} = (x_1, x_2, \ldots, x_N)$ as input, we first mask a fraction of the input tokens and then train the model to predict the masked tokens based on the past and future context. Crucially, when predicting a masked token at position $i$, we compute the loss based on the logits obtained from the token representation at the previous position $i - 1$, not the masked position itself (see Figure 1).

**Unsupervised contrastive learning** While the previous two steps of the LLM2Vec recipe can transform any decoder-only LLM into an encoder for word-level tasks, they might not be sufficient for sequence representations. Unlike bidirectional encoders that include a next sentence prediction objective in their pre-training objectives (Devlin et al., 2019), decoder-only LLMs are not explicitly trained to capture the context of the entire sequence. To fill this gap, we apply unsupervised contrastive learning via SimCSE (Gao et al., 2021). Specifically, given an input sentence, it is passed through the model twice with independently sampled dropout masks, resulting in two different representations for the same sentence. The model is trained to maximize the similarity between these two representations while minimizing the similarity with representations of other sentences in the batch. Crucially, this step does not require any sentence pair data and can be applied using any collection of sentences. We use a pooling operation on the word representations to get the sentence representation (more details in Section 3.2).

### 2.2 Transforming decoder-only LLMs with LLM2Vec

**Models** For most of our results, we experiment with 3 different decoder-only LLMs ranging from 1.3B to 7B parameters: Sheared-LLaMA-1.3B (`S-LLaMA-1.3B`, Xia et al., 2023), Llama-2-7B-chat (`LLaMA-2-7B`, Touvron et al., 2023), and Mistral-7B-Instruct-v0.2 (`Mistral-7B`, Jiang et al., 2023a). In Tables 1 and 2, we provide additional results for the recently released Meta-Llama-3-8B-Instruct model (`Meta-LLaMA-3-8B`, AI@Meta, 2024).

**Training data** We perform both the MNTP and the unsupervised SimCSE step using data from English Wikipedia. We select data from Wikipedia as it is presumably included in the pre-training mixture of all the models we experiment with. It is therefore fair to assume that these two adaptation steps are not teaching the model any new knowledge beyond how to attend to future tokens and how to construct sequence representations. Specifically, we use the Wikitext-103 dataset (Merity et al., 2017) for the MNTP step and a subset of Wikipedia sentences released by Gao et al. (2021) for the unsupervised SimCSE step.

**Masked next token prediction** We follow established practice from the masked language modeling literature and randomly mask a fraction of the tokens from the input sequence (Devlin et al., 2019; Liu et al., 2019). We use the underscore (_) as the mask token, since the models we experiment with do not have a special token for masking. We fine-tune the model using LoRA (Hu et al., 2022) to predict the masked token using the representation

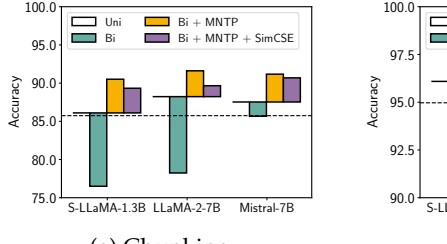 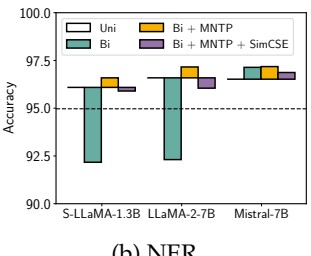 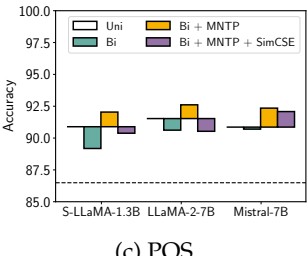

|  |  |  |
|:---:|:---:|:---:|
| (a) Chunking | (b) NER | (c) POS |

Figure 2: Evaluation of LLM2Vec-transformed models on word-level tasks. Solid and dashed horizontal lines show the performance of `Uni` and `DeBERTa-v3-large`, respectively.

of the previous token to maximally align our training objective with the pre-training setup of decoder-only LLMs. For all models, we trained for 1000 steps with a batch size of 32 on a single 80GB A100 GPU. For 7B and 8B models, this training takes only 100 minutes. We provide additional details of our training setup and hyperparameters in Appendix D.1.1.

**Unsupervised contrastive learning** For the contrastive training, we apply the unsupervised SimCSE approach by Gao et al. (2021). The positive examples are constructed by applying LLM's dropout twice on the same input sequence, whereas the other sequences in the batch act as in-batch negatives. We merge the MNTP LoRA weights into the base model and initialize new LoRA parameters before starting the SimCSE training, which ensures that the models retains the knowledge learned in the previous step. Similar to the MNTP step, we train for 1000 steps. For 7B and 8B models, this training takes 3 hours on a single 80GB A100 GPU with a batch size of 128. We provide additional details of our training setup and hyperparameters in Appendix D.1.2.

## 3 LLM2Vec-transformed models are strong unsupervised text embedders

### 3.1 Evaluation on word-level tasks

We start by evaluating on word-level tasks to demonstrate that LLM2Vec is successful at improving the contextual representations constructed by decoder-only LLMs.

**Setup** We evaluate three word-level tasks: chunking, named-entity recognition (NER), and part-of-speech tagging (POS), using the CoNLL-2003 benchmark (Tjong Kim Sang & De Meulder, 2003). We embed each input sentence and train a task-specific linear classifier on top of the frozen representations. This is akin to the linear probing setup commonly used in the language model analysis literature (Belinkov, 2022). We compare the LLM2Vec-transformed models to `DeBERTa-v3-large` (He et al., 2023), the current state-of-the-art encoder-only model. Additional details about our setup are provided in Appendix D.1.3.

**Results** Figure 2 shows the results of our evaluation (a detailed breakdown of the results is provided in Table 4). On each of the three tasks, constructing token representations with causal attention (Uni) already outperforms the encoder-only baseline. This is not surprising, given that the models we experiment with are significantly larger and have been pre-trained on more data. As expected, naively applying bidirectional attention dramatically hurts performance in most cases. Interestingly, for `Mistral-7B`, enabling bidirectional attention hurts performance much less compared to `S-LLaMA-1.3B` and `LLaMA-2-7B`. For NER, Mistral's performance even improves by 0.6% with bidirectional connections.

Focusing on the LLM2Vec-transformed models, we observe that for all models and tasks, adapting via MNTP improves performance. For instance, in the chunking task, we see improvements for `S-LLaMA-1.3B` (by 5%), `LLaMA-2-7B` (by 4%), and `Mistral-7B` (by 4%). Combining MNTP with SimCSE, however, performs worse than just applying MNTP. This is expected for word-level tasks, as SimCSE adapts the representations for sequence-level tasks.

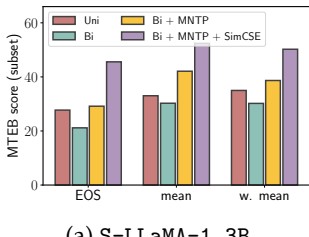 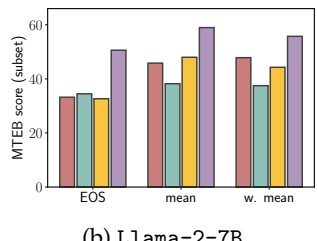 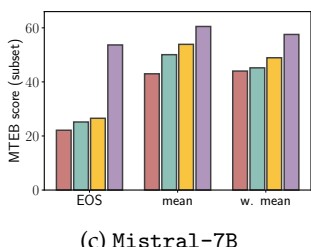

(a) `S-LLaMA-1.3B`           (b) `Llama-2-7B`           (c) `Mistral-7B`

Figure 3: Unsupervised results on our 15 task subset of the MTEB dataset. We ablate three different pooling choices: EOS, mean pooling, and weighted mean pooling. LLM2Vec is compatible with all three approaches and works best with mean pooling.

## 3.2   Evaluation on sequence-level tasks

Next, we evaluate on the Massive Text Embedding Benchmark (MTEB), a collection of 7 diverse embedding task categories covering a total of 56 datasets (Muennighoff et al., 2023). To select the best-performing pooling method for each method, we perform ablations on a 15 task subset consisting of representative tasks from each of the MTEB categories. We provide additional details and justification for how we chose this subset in Appendix C.1.

**Setup**   Following previous work (Su et al., 2023; Wang et al., 2023; Springer et al., 2024), we evaluate with task-specific instructions. For a fair comparison, we use the same set of instructions as Wang et al. (2023) which are also used by Springer et al. (2024). The instructions are only added to queries and can be found in Table 10 of Appendix C.2. For symmetric tasks, the same instruction will be used for the query and the document. When applying (weighted) mean pooling (Muennighoff, 2022), we exclude the instruction tokens.

As a baseline, we compare to the unsupervised BERT models obtained from Gao et al. (2021). Additionally, we compare to Echo embeddings, a concurrent approach by Springer et al. (2024), which we run with the same models and instructions (see Appendix E.1 for more details on our implementation of Echo embeddings). Echo duplicates the input and takes the pooling over the second occurrence to address the limitation of causal information flow.

**Results on our 15 task subset of MTEB**   Figure 3 shows the impact of various pooling methods for all three models on the subset of MTEB tasks. We can clearly observe that applying causal attention is sub-optimal when constructing text embeddings. The dominant paradigm of applying the EOS pooling for models with causal attention is outperformed by (weighted) mean pooling. Enabling bidirectional attention without any training harms performance for `S-LLaMA-1.3B` and `LLaMA-2-7B`. Similar to our word-level results, the performance of `Mistral-7B` improves with bidirectional attention, even without any training.

For LLM2Vec-transformed models, applying MNTP training improves the performance of `S-LLaMA-1.3B` and `Mistral-7B`. Moreover, applying SimCSE further boosts the performance of `S-LLaMA-1.3B`, `LLaMA-2-7B`, and `Mistral-7B` by 49.8%, 23.2%, and 37.5% compared to the best causal baseline on the MTEB subset. We further conduct an ablation of each component of LLM2Vec recipe in Appendix D.2.2 (Table 5).

**Results on full MTEB**   Table 1 shows the results of the best performing models, which we select based on the ablation above, on the full MTEB dataset. After the first two steps of LLM2Vec—bidirectional attention and MNTP—we observe a considerable improvement in performance for all four models (e.g., 16.4% improvement for `Mistral-7B`).

When comparing to Echo embeddings, LLM2Vec (the first two steps only)[3] leads to improved performance for `S-LLaMA-1.3B`, `LLaMA-2-7B`, and `Meta-LLaMA-3-8B`, and performs almost on par for `Mistral-7B`. However, compared to Echo embeddings, LLM2Vec is much

---

[3]We only directly compare the performance after the first two steps of LLM2Vec to Echo embeddings as applying SimCSE involves learning sequence representation, which makes the comparison unfair.

| Categories → | Retr. | Rerank. | Clust. | PairClass. | Class. | STS | Summ. | Avg |
|---|---|---|---|---|---|---|---|---|
| # of datasets → | 15 | 4 | 11 | 3 | 12 | 10 | 1 | 56 |
| Encoder-only | | | | | | | | |
| BERT | 10.59 | 43.44 | 30.12 | 56.33 | 61.66 | 54.36 | 29.82 | 38.33 |
| BERT + SimCSE | 20.29 | 46.47 | 29.04 | 70.33 | 62.50 | 74.33 | 31.15 | 45.45 |
| S-LLaMA-1.3B | | | | | | | | |
| Uni + w. Mean | 9.47 | 38.02 | 28.02 | 42.19 | 59.79 | 49.15 | 24.98 | 35.05 |
| LLM2Vec (w/o SimCSE) | 15.48 | 40.99 | 31.83 | 50.63 | 64.54 | 62.06 | 26.82 | 41.43 |
| LLM2Vec | 25.93 | 47.70 | 37.45 | 72.21 | 67.67 | 71.61 | 31.23 | 49.42 |
| Echo | 10.36 | 41.52 | 30.03 | 52.08 | 63.75 | 59.36 | 22.79 | 39.10 |
| LLaMA-2-7B | | | | | | | | |
| Ui + w. Mean | 15.16 | 46.94 | 36.85 | 61.41 | 69.05 | 63.42 | 26.64 | 44.54 |
| LLM2Vec (w/o SimCSE) | 19.86 | 44.74 | 35.31 | 61.60 | 67.94 | 66.74 | 26.83 | 45.70 |
| LLM2Vec | 36.75 | 52.95 | 40.83 | 77.89 | 71.57 | 76.41 | 31.38 | 55.36 |
| Echo | 16.16 | 46.84 | 34.25 | 63.54 | 69.82 | 67.95 | 25.57 | 45.36 |
| Mistral-7B | | | | | | | | |
| Uni + w. Mean | 10.43 | 45.11 | 35.82 | 60.28 | 71.14 | 58.59 | 26.57 | 42.46 |
| Bi + Mean | 15.84 | 47.40 | 35.55 | 66.53 | 72.18 | 71.04 | 29.93 | 46.86 |
| LLM2Vec (w/o SimCSE) | 19.74 | 50.43 | 40.06 | 70.95 | 72.51 | 71.90 | 27.84 | 49.43 |
| LLM2Vec | 38.05 | **53.99** | 40.63 | **80.94** | **74.07** | **78.50** | 30.19 | **56.80** |
| Echo | 22.68 | 51.07 | 36.78 | 75.87 | 72.69 | 73.60 | 29.54 | 50.26 |
| Meta-LLaMA-3-8B | | | | | | | | |
| Uni + w. Mean | 15.17 | 46.22 | 36.84 | 60.94 | 67.41 | 62.80 | 25.51 | 43.98 |
| Bi + Mean | 3.90 | 34.56 | 14.27 | 42.71 | 57.89 | 51.15 | 23.26 | 30.56 |
| LLM2Vec (w/o SimCSE) | 24.75 | 49.20 | 39.74 | 65.91 | 69.00 | 67.85 | 25.59 | 48.84 |
| LLM2Vec | **39.19** | 53.09 | **41.99** | 78.01 | 71.88 | 75.86 | **31.45** | 56.23 |
| Echo | 12.58 | 49.79 | 36.32 | 68.95 | 70.22 | 67.43 | 26.44 | 45.32 |

Table 1: Unsupervised results on MTEB. We compare `S-LLaMA-1.3B`, `LLaMA-2-7B`, `Mistral-7B`, and `Meta-LLaMA-3-8B` with and without LLM2Vec to the unsupervised BERT models of Gao et al. (2021) as well as Echo embeddings (Springer et al., 2024).

more efficient as Echo embeddings repeat the input and therefore double the sequence length which makes inference considerably slower (we provide a runtime comparison in Appendix E.2). Adding the final step of the LLM2Vec recipe—unsupervised SimCSE—further boosts all three models by a large margin, making our LLM2Vec `Mistral-7B` SOTA among all unsupervised models with the score of 56.80.

Interestingly, `Meta-LLaMA-3-8B` with LLM2Vec (w/o SimCSE) outperforms echo embeddings by a larger margin compared to the other models. Adding SimCSE again boosts performance, but does not outperform LLM2Vec applied to `Mistral-7B`.

Overall, our results highlight that LLM2Vec is successful at transforming decoder-only LLMs into strong text embedding models which outperform previous unsupervised approaches on the challenging MTEB leaderboard.

## 4 How does LLM2Vec affect a model?

### 4.1 LLM2Vec helps models to capture information from future tokens

To analyze the extent to which LLM2Vec-transformed models incorporate information from future tokens, we adopt the analysis of Springer et al. (2024) and test how well the model performs at judging the similarity between sentences that share the same prefix.

**Setup** We evaluate on a synthetic dataset collected by Springer et al. (2024), which consists of 35 sentence triples $\{(q_i, s_i^+, s_i^-)\}_{i=1}^{35}$ with $q_i = (A_i, B_i)$, $s_i^+ = (A_i, C_i)$, and $s_i^- = (A_i, D_i)$, where $B_i$ and $C_i$ have a similar meaning but $B_i$ and $D_i$ don't. We compute a sequence

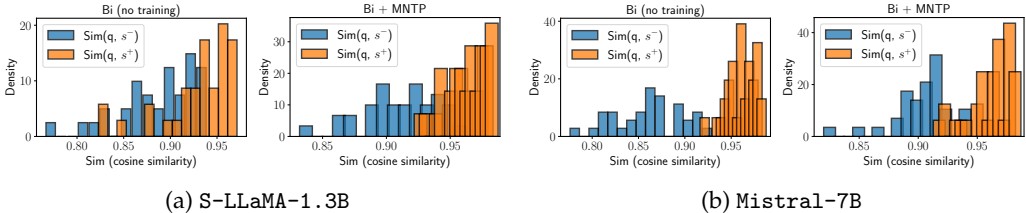

(a) S-LLaMA-1.3B         (b) Mistral-7B

Figure 4: Cosine similarity between query ($q$) and negative ($s^-$) as well as positive examples ($s^+$). Plots for LLaMA-2-7B and other approaches are shown in Appendix F.

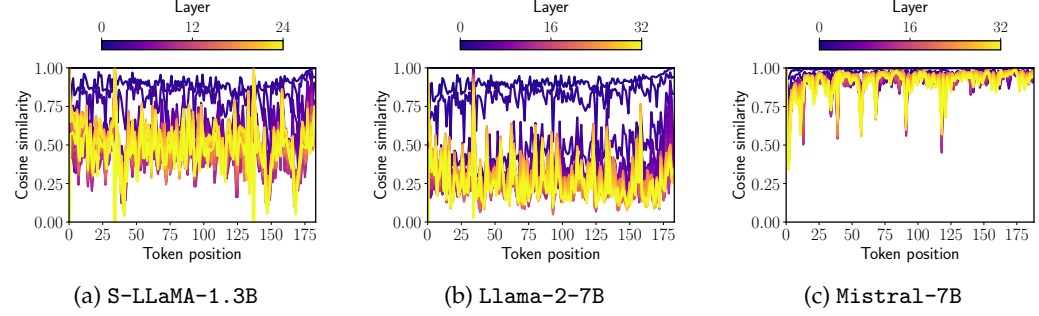

(a) S-LLaMA-1.3B     (b) Llama-2-7B     (c) Mistral-7B

Figure 5: Cosine similarities at different token positions at layers when comparing representations constructed with causal attention to those constructed with bidirectional attention (without training). Additional plots are shown in Appendix F.

representation for each of these sentences by pooling only over the first part of the sentence, i.e., $A_i$. We then compute the cosine similarity between the resulting embeddings. A model that incorporates information from future tokens ($B_i$, $C_i$, or $D_i$) in the representations of the prefix $A_i$ should assign a higher similarity to the positive example.

**Results** Figure 4 shows the results of our analysis for S-LLaMA-1.3B and Mistral-7B. Results for LLaMA-2-7B, which show the same trends, and a comparison to Echo embeddings are provided in Appendix F. For S-LLaMA-1.3B, we observe that enabling bidirectional attention and training with the MNTP objective are sufficient to establish a clear separation between the positive and negative examples. For Mistral-7B, all setups lead to a larger cosine similarity between the query and positive than the query and negative examples.

### 4.2 Why does bidirectional attention without training work for Mistral models?

Our empirical results so far as well as the analysis above share an intriguing observation: enabling bidirectional attention works well for Mistral-7B, even without any training. Below, we investigate this surprising behavior by analyzing how bidirectional attention impacts the representations of a model.

**Setup** We feed a single input sequence (a random paragraph from Wikipedia) to each model and compute the hidden representations of every token at every layer $l$ with causal ($\mathbf{H}_l^c$) and bidirectional attention ($\mathbf{H}_l^{bi}$). For every layer, we compute the cosine similarity between the representations constructed using causal and bidirectional attention, i.e., $\text{sim}(\mathbf{H}_l^c, \mathbf{H}_l^{bi})$. For most layers, we expect this similarity to be low, as enabling bidirectional attention without any training should lead to substantially different representations.

**Results** Figure 5 shows that as expected, for S-LLaMA-1.3B and LLaMA-2-7B, enabling bidirectional attention without training has a profound impact on the representations, leading to low cosine similarity across almost all layers and token positions. For Mistral-7B, on the other hand, the representations have very high cosine similarity throughout.

| Categories → | Retr. | Rerank. | Clust. | PairClass. | Class. | STS | Summ. | Avg |
|---|---|---|---|---|---|---|---|---|
| # of datasets → | 15 | 4 | 11 | 3 | 12 | 10 | 1 | 56 |
| Previous work w/ public data only | | | | | | | | |
| Instructor-xl | 49.26 | 57.29 | 44.74 | 86.62 | 73.12 | 83.06 | **32.32** | 61.79 |
| BGElarge-en-v1.5 | 54.29 | 60.03 | 46.08 | 87.12 | 75.97 | 83.11 | 31.61 | 64.23 |
| GritLMMistral-7b-v1 + public data | 53.10 | **61.30** | **48.90** | 86.90 | 77.00 | 82.80 | 29.40 | 64.70 |
| E5Mistral-7b-v1 + public data | 52.78 | 60.38 | 47.78 | **88.47** | 76.80 | 83.77 | 31.90 | 64.56 |
| EchoMistral-7b-v1 | 55.52 | 58.14 | 46.32 | 87.34 | **77.43** | 82.56 | 30.73 | 64.68 |
| S-LLaMA-1.3B | | | | | | | | |
| Uni + w. Mean | 51.02 | 54.65 | 39.90 | 83.57 | 71.64 | 82.16 | 30.05 | 60.44 |
| LLM2Vec (w/o SimCSE) | 51.44 | 55.38 | 43.57 | 86.20 | 72.21 | 83.58 | 30.01 | 61.85 |
| LLM2Vec | 51.49 | 55.58 | 43.24 | 85.80 | 72.98 | 83.62 | 30.12 | 61.96 |
| LLaMA-2-7B | | | | | | | | |
| Uni + w. Mean | 54.33 | 58.01 | 40.57 | 87.01 | 75.60 | 83.47 | 29.68 | 62.96 |
| LLM2Vec (w/o SimCSE) | 54.60 | 57.38 | 45.24 | 88.03 | 76.33 | 83.73 | 28.49 | 64.14 |
| LLM2Vec | 54.34 | 57.70 | 45.04 | 87.87 | 76.53 | 83.43 | 28.82 | 64.04 |
| Mistral-7B | | | | | | | | |
| Uni + w. Mean | 54.81 | 57.37 | 41.07 | 86.05 | 76.01 | 83.44 | 30.74 | 63.20 |
| LLM2Vec (w/o SimCSE) | 55.99 | 58.42 | 45.54 | 87.99 | 76.63 | **84.09** | 29.96 | **64.80** |
| LLM2Vec | 56.05 | 58.59 | 45.12 | 88.18 | 76.72 | 83.69 | 30.66 | 64.72 |
| Meta-LLaMA-3-8B | | | | | | | | |
| Uni + w. Mean | 55.42 | 58.60 | 43.19 | 86.29 | 75.56 | 83.95 | 30.59 | 63.87 |
| LLM2Vec (w/o SimCSE) | 56.63 | 59.68 | 46.45 | 87.80 | 75.92 | 83.58 | 30.94 | **65.01** |
| LLM2Vec | **56.71** | 59.02 | 45.86 | 87.95 | 76.67 | 82.98 | 29.67 | 64.90 |

Table 2: Supervised results on full MTEB benchmark. The best performing LLM2Vec model `Meta-LLaMA-3-8B` + LLM2Vec (w/o SimCSE) achieves a new SOTA performance among models trained only on publicly available data.

Based on these findings (we replicate these results for other inputs and other variants of Mistral in Appendix F) and the strong unsupervised results for `Mistral-7B` with bidirectional attention, we speculate that Mistral models are pre-trained with some form bidirectional attention, e.g., prefix language modeling (Raffel et al., 2020) – at least for some parts of its training. We leave a more detailed investigation of this intriguing behavior for future work.

# 5 Combining LLM2Vec with supervised contrastive learning

The final piece of our evaluation combines LLM2Vec with supervised contrastive learning.

## 5.1 LLM2Vec leads to strong performance on the MTEB leaderboard

**Setup** For supervised training, we train on a replication of the public portion of the E5 dataset (Wang et al., 2023) curated by Springer et al. (2024). The dataset consists of approximately 1.5M samples and we provide details on its compilation in Appendix G.1. We follow standard practice and train the models with contrastive learning using hard negatives and in-batch negatives. We use LoRA fine-tuning for supervised setting as well. The MNTP LoRA weights are merged into the base model, and the trainable LoRA weights are initialized with SimCSE weights. For LLM2Vec models that use just MNTP, the LoRA weights are randomly initialized. The training is performed for 1000 steps with a batch size of 512. We detail other hyperparameters in Appendix G.2.

For a fair comparison, we only compare to models trained on publicly available data and provide a comparison to the top entries on the MTEB leaderboard in Appendix G.3.

**Results** Table 2 shows the results of our evaluation. For all models, transforming a model with LLM2Vec leads to improved performance over the strong `Uni` + weighted mean baseline. As expected, performing unsupervised SimCSE is less crucial for supervised training, and even leads to slightly worse performance for `LLaMA-2-7B`, `Mistral-7B`, and

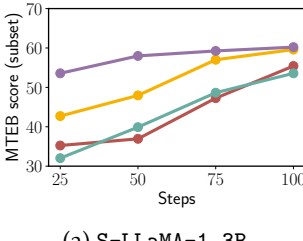 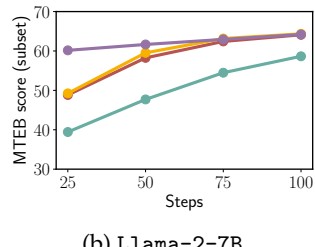 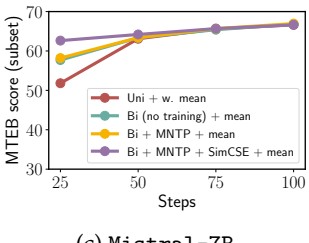

(a) `S-LLaMA-1.3B`  (b) `Llama-2-7B`  (c) `Mistral-7B`

Figure 6: Results on the 15 task subset of MTEB during training of `S-LLaMA-1.3B`, `LLaMA-2-7B`, and `Mistral-7B`. For all three models, applying LLM2Vec before supervised training leads to better performance with less steps.

`Meta-LLaMA-3-8B` compared to just performing the MNTP step of LLM2Vec (LLM2Vec w/o SimCSE). However, as we will show in Section 5.2, LLM2Vec with MNTP and SimCSE is much more sample-efficient, and therefore crucial in compute or data-constrained settings. Notably, our best model, `Meta-LLaMA-3-8B` + LLM2Vec (w/o SimCSE) leads to a new state-of-the-art performance among models trained only on publicly available data.

### 5.2 LLM2Vec leads to more sample-efficient training

**Setup**   To demonstrate the sample-efficiency of LLM2Vec-transformed models, we save a checkpoint every 25 training steps and evaluate them on our 15 task subset of MTEB.

**Results**   As shown in Figure 6, LLM2Vec-transformed models reach better performance earlier in training. This observation is consistent across all three models. For `S-LLaMA-1.3B`, the smallest of our three models, even performing just MNTP leads to a considerably improved sample-efficiency. These results are particularly encouraging for settings where it is hard to acquire high quality labeled data, a setting which we leave for future work.

## 6   Related Work

**Supervised text encoders**   Initially, supervised methods primarily relied on tasks such as natural language inference or sentence similarity to train BERT-like models for producing sentence embeddings (Conneau et al., 2017; Reimers & Gurevych, 2019). Subsequently, BERT-like models have also been adapted to tasks like retrieval (Karpukhin et al., 2020; Khattab & Zaharia, 2020). More recent methods have further improved these representations through a complex multi-stage learning pipeline that consists of large-scale weakly supervised contrastive training followed by multi-task fine-tuning (Ni et al., 2022; Wang et al., 2022a; Li et al., 2023a; Xiao et al., 2023) Recent approaches have focused on enhancing the generalization and transferability of text embeddings using instructions (Su et al., 2023; Asai et al., 2023).

**Unsupervised text encoders**   Another line of work has explored training text embedders in an unsupervised manner using only a set of unordered sentences. These unsupervised approaches typically create two different representations of the same sentence for contrastive learning. The methods vary in how they form these representations – perturbing the input sentence (Wu et al., 2020), or using different model instances (Carlsson et al., 2021). SimCSE (Gao et al., 2021), the approach used in this work, generates two representations of the same sentence by passing it through the model twice with different dropout masks.

**Turning decoder-only LLMs into text encoders**   While decoder-only LLMs have outperformed bidirectional encoders across a large variety of language understanding tasks (Brown et al., 2020; Touvron et al., 2023; Jiang et al., 2023a, *inter alia*), their impact on sentence representation learning remains limited. The most common approaches in literature use the final hidden state of the last token as the sentence embedding (Neelakantan et al., 2022; Ma et al., 2023; Wang et al., 2023).

There are few works that explore the limitations of using a causal attention mask when adapting decoder-only LLMs for text classification and sentence representation tasks. Li et al. (2023b) experiment with removing the causal mask of Llama-2 during supervised fine-tuning for text classification and NER tasks. Similarly, Dukić & Šnajder (2024) enable bidirectional attention for a group of layers during supervised fine-tuning on NER and chunking. In the context of sentence representation learning, Li & Li (2024) explore enabling bidirectional attention in the last layer of a decoder-only model during supervised contrastive fine-tuning on STS tasks.

Concurrent to our work, several works have focused on converting decoder-only-LLMs to text encoders in supervised and unsupervised manner. Jiang et al. (2023b) and Lei et al. (2024) prompt the language model to summarize the input text in one word, and take the last layer's hidden embedding for the last token as the text's representation. Muennighoff et al. (2024) perform multi-task full fine-tuning using a combination of self-supervised language modeling with causal attention and supervised contrastive learning with bidirectional attention. In contrast, our proposed approach is much more computationally efficient, as it requires only parameter-efficient fine-tuning and 1000 gradient steps. Closest to our work is the concurrent work of Springer et al. (2024). They propose to copy the input sequence and append it to itself, which addresses the contextualization issue of causal attention as tokens in the copy of the input can now attend to "future" tokens in the previous sequence. While this performs well in practice, it significantly increases the computational cost at inference time, which can be particularly problematic for encoding longer documents. Our approach outperforms Springer et al. (2024), without inducing any additional computational overhead at inference time.

## 7    Conclusion

We present LLM2Vec, a strong unsupervised approach to transform any decoder-only LLMs into a (universal) text embedder. We perform an extensive evaluation on word- and sequence-level tasks and demonstrate the effectiveness of LLM2Vec in both unsupervised and supervised settings. Applying LLM2Vec to `Mistral-7B` achieves a new state-of-the-art performance on MTEB among unsupervised approaches. When combining LLM2Vec with supervised contrastive fine-tuning, `Meta-LLaMA-3-8B` achieves SOTA performance among approaches that train only on publicly available data (as of May 24, 2024). Beyond our strong empirical contributions, we provide an extensive analysis of how LLM2Vec impacts the underlying model and reveal an intriguing property of `Mistral-7B`, which explains its strong out of the box performance with bidirectional attention. The simplicity of our approach, as well as its compute and sample-efficiency, makes LLM2vec a promising solution for low-resource and compute constrained scenarios and opens up several interesting avenues for future work.

## Acknowledgements

We thank the members of SR's research group for providing feedback throughout the project. Furthermore, we thank Jacob Mitchell Springer for providing the supervised training data used in Springer et al. (2024). PB is supported by the Mila-Intel Grant program. MM is partly funded by the Mila P2v5 Technology Maturation Grant and the Mila-Samsung grant. SR is supported by a Facebook CIFAR AI Chair and NSERC Discovery Grant program.

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

## A   Limitations

**Large size of decoder-only LLMs**   Recent years have seen a increasing trend towards training very large decoder-only LLMs, with model sizes up to 540B parameters (Brown et al., 2020; Chowdhery et al., 2023). The parameter size of the model has a direct impact on the training and inference latency. Additionally, the large output embedding dimension of these models (e.g., 4096 for `Mistral-7B` compared to 768 for `BERT`) also makes them more memory and compute intensive for creating vector indexes for large document collections. While some of these limitations can be offset by recent advances in improving the training and inference efficiency of large models (Hu et al., 2022; Dao, 2024), these techniques can technically be applied to smaller bidirectional models as well.

The advantages of small bidirectional encoders come at the cost of complex and computationally intensive training regimes (Li et al., 2023a; Xiao et al., 2023; Li et al., 2023a). In contrast, decoder-only models are much more sample-efficient and do not require large-scale contrastive pre-training (Wang et al., 2023). Moreover, the instruction following capabilities of decoder-only models make them strong contenders for building text embedding models that generalize to a wide range of tasks and domains without the need for expensive adaptation.

While smaller models can be more practical for some applications, the sample-efficiency, the instruction following capabilities, and the widespread use of these models in the community motivates the need to explore the potential of decoder-only LLMs for text embedding tasks.

**Data contamination from pre-training**   As our supervised data contains train splits of publicly available datasets, there is an extremely low chance of test set contamination with the MTEB benchmark. However there is a possibility of contamination from the pre-training data of `LLaMA-2-7B` and `Mistral-7B` models (`S-LLaMA-1.3B` was distilled from `LLaMA-2-7B`). As the complete details of the pre-training data are not publicly available, we cannot be certain about the extent of contamination. However, to reliably compare with other works, we stick to our choice of model and evaluation benchmark. We leave it to future work to investigate the performance of these models on newly designed benchmarks that are not part of their pre-training data.

**Extending to other languages**   In this work, we have implemented and evaluated our proposed methodology – LLM2Vec – using only English text corpora and benchmarks. However, the methodology is language-agnostic and can be easily extended to other languages using just unstructured text collections. We leave it to future work to investigate the performance of LLM2Vec on other languages.

## B   Background

### B.1   Self-attention

The self-attention mechanism is a crucial component of decoder-only LLMs Cheng et al. (2016); Lin et al. (2017); Vaswani et al. (2017); Paulus et al. (2018). Given a sequence of $N$ tokens, the token representation at any given transformer layer $(\mathbf{x}_1, \mathbf{x}_2, \ldots, \mathbf{x}_N)$ with $\mathbf{x}_i \in \mathbf{R}^d$ are stacked into a matrix $\mathbf{X} \in \mathcal{R}^{N \times d}$. Given this matrix, the self-attention mechanism computes the query, key, and value matrices $\mathbf{Q}, \mathbf{K}, \mathbf{V} \in \mathcal{R}^{N \times p}$ via a learned linear transformation.

$$\mathbf{Q} = X\mathbf{W}^Q, \tag{1}$$
$$\mathbf{K} = X\mathbf{W}^K, \tag{2}$$
$$\mathbf{V} = X\mathbf{W}^V. \tag{3}$$

The output of the self-attention layer is then computed as a linear combination of the values, weighted by the normalized inner product between keys and queries:

$$\mathbf{O} = \text{softmax}\left(\frac{\mathcal{M}_{\{j \leq i\}}\mathbf{Q}\mathbf{K}^T}{\sqrt{d}}\right)\mathbf{V}. \tag{4}$$

This output is then passed through a feed-forward network and added to the residual stream to obtain the token representations at the next layer. Crucially, in the case of decoder-only LLMs, the attention mask $\mathcal{M}_{\{j \leq i\}}$ prevents accessing token embeddings to the right of the current token.

## B.2   Contrastive learning

Contrastive learning is a popular paradigm to learn text representations Karpukhin et al. (2020); Gao et al. (2021); Su et al. (2023); Wang et al. (2023); Springer et al. (2024). In the supervised setup, we have a set of positive pairs $\mathcal{D} = \{(q_i, d_i^+)\}_{i=1}^n$, and a set of negative documents that can include hard or in-batch negatives. The model is trained to maximize the similarity (i.e., usually cosine similarity) of positive pairs and minimize the similarity of negative pairs, i.e., we optimize the following objective:

$$\mathcal{L} = \frac{e^{\lambda s(q, d^+)}}{e^{\lambda s(q, d^+)} + \sum_{d^- \in N} e^{\lambda s(q, d^-)}}, \tag{5}$$

where $s$ is a similarity metric, $\lambda$ a temperature value, and $N$ all the negative documents for query $q$.

**Unsupervised contrastive learning**   In unsupervised contrastive learning, no positive or hard negative pairs are available. Most unsupervised approaches construct two different representation for the same sample, using either model or input perturbations. SimCSE (Gao et al., 2021), the unsupervised approach used in this work, creates two different representations of the same input by using independently sampled dropout masks in the intermediate model representations and train the model with in-batch negatives.

## C   Massive Text Embeddings Benchmark (MTEB)

### C.1   MTEB subset details

MTEB consists of diverse small and large embedding tasks. To speed up the evaluation[4], we consider a representative subset of 15 tasks from MTEB for our analyses, presented in Table 3. To make sure that our ablation and analyses are not biased towards one specific category or task, this subset includes tasks from each category with almost the same proportion compared to the full MTEB[5].

### C.2   MTEB instructions

When evaluating on MTEB, we use the same instructions as Wang et al. (2023). The list of instructions for each task is listed in Table 10.

---

[4]Full evaluation on MTEB takes more than 40h for `Mistral-7B` on 8x A100 GPUs.

[5]Since the MTEB's SummEval category includes only one dataset, we skip this category in our small-scale evaluation.

| Category | Dataset |
|---|---|
| Retrieval (3) | SciFact
ArguAna
NFCorpus |
| Reranking (2) | StackOverflowDupQuestions
SciDocsRR |
| Clustering (3) | BiorxivClusteringS2S
MedrxivClusteringS2S
TwentyNewsgroupsClustering |
| Pair Classification (1) | SprintDuplicateQuestions |
| Classification (3) | Banking77Classification
EmotionClassification
MassiveIntentClassification |
| STS (3) | STS17
SICK-R
STSBenchmark |
| SummEval (0) | - |
| Overall | 15 datasets |

Table 3: Subset of MTEB tasks used for our ablations and analysis.

# D   Details on unsupervised results

## D.1   Training details

### D.1.1   MNTP training details

The second step of LLM2Vec includes MNTP training. We follow established practice from the encoder-only literature for choosing our masking strategy. For example, (Devlin et al., 2019) mask 15% of the tokens in the input. 10% of the masked tokens are then replaced with a random token from the vocabulary, while another 10% are unmasked again, but still considered when computing the loss. As another example, RoBERTa (Liu et al., 2019) also masks 15% of the input tokens but applies no further post-processing to the masked tokens.

For our models, we perform a hyperparameter search to select the percentage of the masked tokens in a sequence choosing from 20%, 40%, 60%, 80%, and 90%. For each model, we take the best setup (i.e., masking probability and BERT vs. RoBERTa approach) based on the performance on SICK-R (Agirre et al., 2014) task from the MTEB dataset. This results in the following choices: for `S-LLaMA-1.3B`, `LLaMA-2-7B`, and `Meta-LLaMA-3-8B`, we apply BERT's masking strategy with masking probability of 20%. For `Mistral-7B`, we apply RoBERTa's masking strategy with probability of 80%.

We train all the models for 1000 steps with LoRA $r = 16$ and $\alpha = 32$, and we follow the same training parameters as RoBERTa MNTP training. When training large 7B and 8B models, we apply brain floating point (bfloat16) quantization, as well as flash attention 2 and gradient checkpointing.

### D.1.2   SimCSE training details

The last step of LLM2Vec involves unsupervised contrastive learning with SimCSE. Our initial experiments indicated that the low value of dropout probability (0.1) typically used by bidirectional encoders (Gao et al., 2021) does not lead to optimal performance for larger decoder-only LLMs. Therefore, we use a higher dropout probability of 0.3 for all models.

Similar to MNTP, we train all models with LoRA $r = 16$ and $\alpha = 32$ for 1000 steps. For `LLaMA-2-7B`, `Mistral-7B`, and `Meta-LLaMA-3-8B`, we train with a batch size of 128. For `S-LLaMA-1.3B`, we use a batch size of 32. Additionally, when training `LLaMA-2-7B`

`Mistral-7B`, and `Meta-LLaMA-3-8B`, we apply brain floating point (bfloat16) quantization, flash attention 2, and gradient checkpointing.

### D.1.3 Word-level training details

We evaluate on three popular word embedding tasks: chunking, named-entity recognition (NER), and part-of-speech (POS) tagging. We train a linear classifier using dropout with a dropout probability of 0.1 on top of the frozen representations obtained from the last hidden layer of a model.

We use data from CoNLL-2003, consisting of roughly 14,000 training, 3,250 validation, and 3,450 test samples (Tjong Kim Sang & De Meulder, 2003). We train the classifier for 1,500 steps with a learning rate of $5e - 4$ and a batch size of 8. For experiments with `Mistral-7B` models that have been tuned with MNTP, we use the variant which is trained with BERT's masking strategy and masking probability of 20% (please see D.1.1 for more details). Although 80% masking helps with the performance in sentence-level tasks, it prevents the model from learning proper token representations essential for word-level tasks.

Since the models we experiment with have sub-token based vocabularies, we calculate the embedding of a word by averaging the representations of all its sub-tokens. For example, for a sentence "$w_1\ w_2\ w_3$" which is tokenized as "BOS $t_{11}t_{12}\ t_{21}t_{22}t_{23}\ t_{31}$", the representation of $w_1$, $w_2$, and $w_3$ will be computed as

$$e_1 = \frac{1}{2}\left(e_{11} + e_{12}\right), \quad e_2 = \frac{1}{3}\left(e_{21} + e_{22} + e_{23}\right), \quad e_3 = e_{31}.$$

Here, $e_.$ is the final representation of token $t_.$ or word $w_.$. Moreover, for the models that have gone through MNTP, we calculate the representation based on sub-tokens of the previous word. Using the same example as above, for models trained with MNTP, the representation of words $w_1$, $w_2$, and $w_3$ will be computed as:

$$e_1 = \frac{1}{2}\left(e_{\text{BOS}} + e_{11}\right), \quad e_2 = \frac{1}{3}\left(e_{12} + e_{21} + e_{22}\right), \quad e_3 = e_{23}.$$

## D.2 Additional results

### D.2.1 Word-level task results

We present the detailed breakdown of the performance of LLM2Vec-transformed models on the word-level tasks in Table 4. Our results show that applying MNTP training to decoder-only LLMs helps them take advantage of the enabled bidirectional attention which boosts their performance on word-level tasks.

### D.2.2 Sentence-level task results

Table 5 presents the results on MTEB subset for all models across different pooling methods. Results show that While weighted mean works the best for causal (i.e., Uni) models, mean pooling performs the best for LLM2Vec approach.

In Table 11, we additionally present a breakdown of the unsupervised performance of LLM2Vec-transformed models on MTEB.

# E Comparison with Echo embedding

## E.1 Reproducibility

Concurrent to our work, Springer et al. (2024) proposed Echo embeddings, a simple approach to convert decoder-only LLMs into text embedders by copying the input sequence and appending it to itself. For evaluation, they follow a prompt sampling procedure for the task instruction. However, they report that the exact wording or template used as a prompting strategy does not have a strong effect on the performance.

| Model | Chunking | NER | POS tagging |
|---|---|---|---|
| `Encoder-only` | | | |
| BERT-large | 71.77 | 90.09 | 75.12 |
| DeBERTa-large | 85.74 | 94.97 | 86.49 |
| `S-LLaMA-1.3B` | | | |
| Uni | 86.10 | 96.09 | 90.89 |
| Bi | 76.50 | 92.17 | 89.18 |
| Bi + MNTP | **90.51** | **96.59** | **92.04** |
| Bi + SimCSE | 75.93 | 91.45 | 89.22 |
| Bi + MNTP + SimCSE | 89.33 | 95.90 | 90.38 |
| `LLaMA-2-7B` | | | |
| Uni | 88.23 | 96.59 | 91.53 |
| Bi | 78.24 | 92.31 | 90.62 |
| Bi + MNTP | **91.61** | **97.16** | **92.61** |
| Bi + SimCSE | 77.75 | 91.96 | 90.48 |
| Bi + MNTP + SimCSE | 89.66 | 96.05 | 90.53 |
| `Mistral-7B` | | | |
| Uni | 87.53 | 96.52 | 90.86 |
| Bi | 85.66 | 97.14 | 90.70 |
| Bi + MNTP | **91.17** | **97.18** | **92.35** |
| Bi + SimCSE | 86.91 | 97.15 | 92.12 |
| Bi + MNTP + SimCSE | 90.69 | 96.87 | 92.08 |

Table 4: Unsupervised results on the word-level tasks for different models.

| Model | EOS | Mean | W. mean |
|---|---|---|---|
| `S-LLaMA-1.3B` | | | |
| Uni | 27.72 | 33.03 | 34.99 |
| Bi | 21.16 | 30.26 | 30.20 |
| Bi + MNTP | 29.16 | 42.10 | 38.67 |
| Uni + SimCSE | 37.44 | 44.95 | 47.13 |
| Bi + SimCSE | 40.43 | 44.46 | 44.83 |
| Bi + MNTP + SimCSE | 45.57 | **52.40** | 50.23 |
| `LLaMA-2-7B` | | | |
| Uni | 33.23 | 45.83 | 47.85 |
| Bi | 34.47 | 38.22 | 37.50 |
| Bi + MNTP | 32.66 | 48.00 | 44.30 |
| Uni + SimCSE | 38.47 | 52.03 | 53.55 |
| Bi + SimCSE | 40.37 | 44.13 | 44.08 |
| Bi + MNTP + SimCSE | 50.61 | **58.97** | 55.75 |
| `Mistral-7B` | | | |
| Uni | 22.12 | 43.00 | 44.01 |
| Bi | 25.17 | 50.07 | 45.20 |
| Bi + MNTP | 26.54 | 53.89 | 48.93 |
| Uni + SimCSE | 34.60 | 52.04 | 53.95 |
| Bi + SimCSE | 49.73 | 60.29 | 56.56 |
| Bi + MNTP + SimCSE | 53.67 | **60.50** | 57.55 |

Table 5: Unsupervised results on MTEB subset for different models.

For a fair comparison to our proposed models, we implement Echo embeddings using the instructions in our evaluation setup (Appendix C.2). To do a sanity check on our implementation, as well as to see the impact of exact wording of instructions, we evaluate

Echo embedding on the same subset of 26 MTEB tasks that was chosen in their work. We run this evaluation using the `Mistral-7B-Instruct-v0.1` model to ensure that the results are directly comparable to theirs.

The unsupervised Echo model based on our implementation and instructions achieved a score of 55.22 on the 26 task subset, whereas their reported score is 55.07. This result validates our implementation and confirms an observation made by Springer et al. (2024) – the exact wording or template used does not have a strong effect on the performance.

### E.2    Efficiency

In Table 6, we report the approximate evaluation time it took (in hours) to evaluate each of the models on MTEB using 8x 80GB A100 GPUs. Given that Echo embeddings rely on copying the input text, evaluation takes much longer compared to our approach. We note that the increased inference time of Echo embeddings is especially problematic for the encoding of large retrieval corpora in MTEB benchmark.

| Model | LLM2Vec | Echo embeddings |
|---|---|---|
| `S-LLaMA-1.3B` | $\approx$ 30 hrs | $\approx$ 40 hrs |
| `LLaMA-2-7B` | $\approx$ 42 hrs | $\approx$ 63 hrs |
| `Mistral-7B` | $\approx$ 44 hrs | $\approx$ 64 hrs |

Table 6: Evaluation time of Echo Embeddings compared to LLM2Vec in hours on 8x 80GB A100 GPUs.

## F    More analysis results

**Data used for our analysis**    Table 7 shows examples of the data used for our cosine similarity analysis in Section 4.

| $q$: **the query** | $s^+$: **the positive sample** | $s^-$: **the negative sample** |
|---|---|---|
| *She loves to travel in summer,* especially to cold destinations, avoiding hot and crowded places. | *She loves to travel in summer,* specifically to chilly locations, steering clear of warm, populous areas. | *She loves to travel in summer,* but prefers to visit hot and bustling tourist spots. |
| *The cat often sits by the window,* dreaming of chasing birds and enjoying the warm sunshine. | *The cat often sits by the window,* imagining bird pursuits and basking in the sunlight. | *The cat often sits by the window,* but is too lazy to dream of chasing anything. |
| *He reads books every night,* finding solace in fiction and escaping from the stresses of daily life. | *He reads books every night,* seeking comfort in stories and evading everyday tensions. | *He reads books every night,* yet he feels that non-fiction is more engaging and informative. |
| *She paints landscapes on weekends,* expressing her creativity through vibrant colors and abstract forms. | *She paints landscapes on weekends,* showcasing her artistic flair with lively hues and unconventional shapes. | *She paints landscapes on weekends,* preferring realistic and detailed depictions of nature. |

Table 7: Toy data used for the analysis in Section 4. These sentences were originally collected by Springer et al. (2024).

**Cosine similarity analysis**    Figure 7 provide cosine similarity results for all three models. In addition to our LLM2Vec-transformed models, we also provide results for Echo emebddings.

## F.1 Additional plots

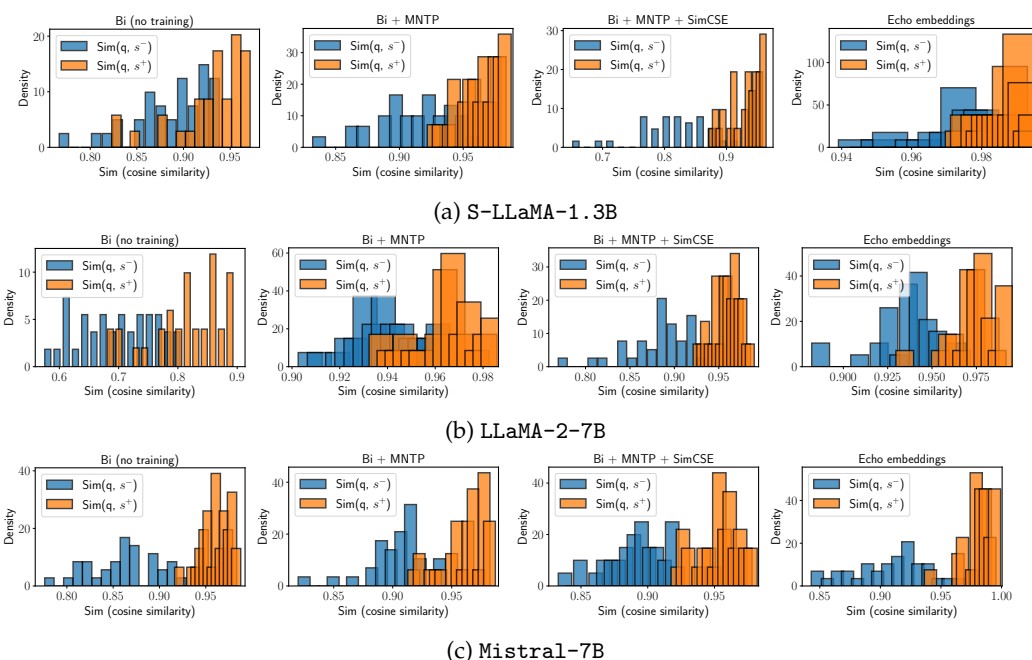

(a) `S-LLaMA-1.3B`

(b) `LLaMA-2-7B`

(c) `Mistral-7B`

Figure 7: Cosine similarity between query and negative as well as positive examples for `S-LLaMA-1.3B`, `LLaMA-2-7B`, and `Mistral-7B`.

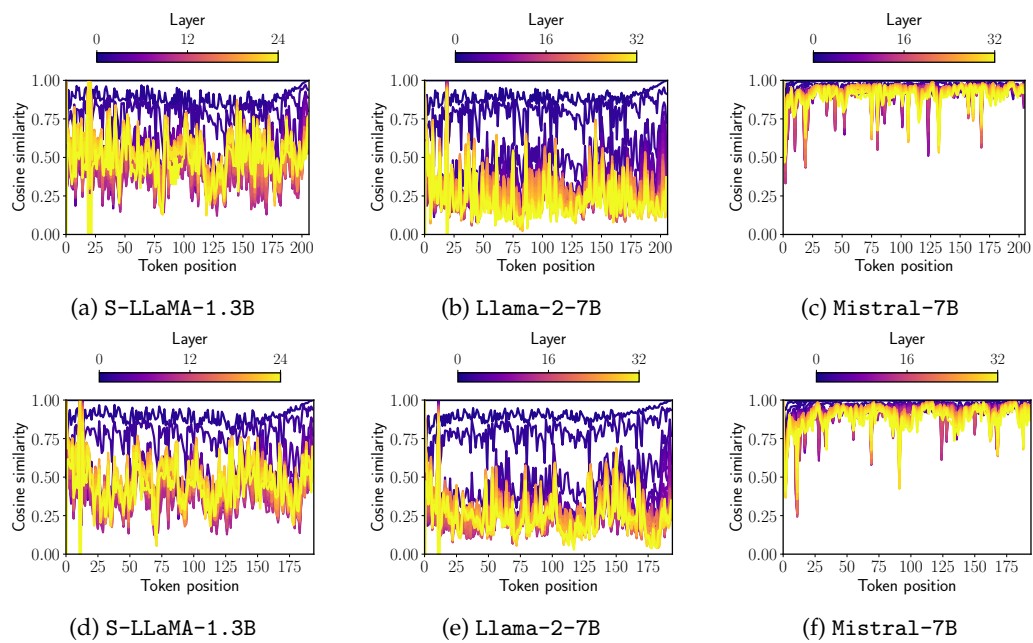

(a) `S-LLaMA-1.3B`    (b) `Llama-2-7B`    (c) `Mistral-7B`

(d) `S-LLaMA-1.3B`    (e) `Llama-2-7B`    (f) `Mistral-7B`

Figure 8: Cosine similarities at different token positions at layers when comparing representations constructed with causal attention to those constructed with bidirectional attention (without training).

**Representation analysis**    Figure 8 provides additional plots for the representation analysis using two different Wikipedia paragraphs. The trends closely follow those reported in Section 4. Figure 9 shows that the same behavior we observe for `Mistral-7B-instruct-v0.2`

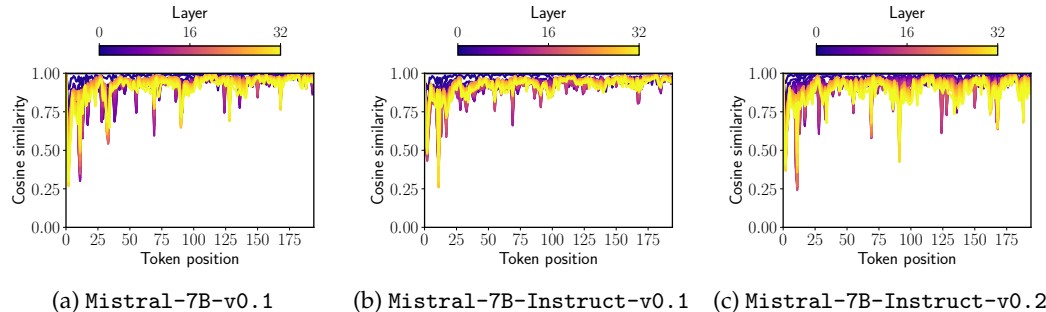

(a) `Mistral-7B-v0.1`    (b) `Mistral-7B-Instruct-v0.1`    (c) `Mistral-7B-Instruct-v0.2`

Figure 9: Cosine similarities at different token positions at layers when comparing representations of Mistral models constructed with causal attention to those constructed with bidirectional attention (without training).

| Dataset | Instruction(s) |
|---|---|
| NLI | Given a premise, retrieve a hypothesis that is entailed by the premise |
| | Retrieve semantically similar text |
| DuReader | Given a Chinese search query, retrieve web passages that answer the question |
| ELI5 | Provided a user question, retrieve the highest voted answers on Reddit ELI5 forum |
| FEVER | Given a claim, retrieve documents that support or refute the claim |
| HotpotQA | Given a multi-hop question, retrieve documents that can help answer the question |
| MIRACL | Given a question, retrieve Wikipedia passages that answer the question |
| MrTyDi | Given a question, retrieve Wikipedia passages that answer the question |
| MSMARCO Passage | Given a web search query, retrieve relevant passages that answer the query |
| MSMARCO Document | Given a web search query, retrieve relevant documents that answer the query |
| NQ | Given a question, retrieve Wikipedia passages that answer the question |
| QuoraDuplicates | Given a question, retrieve questions that are semantically equivalent to the given question |
| | Find questions that have the same meaning as the input question |
| SQuAD | Retrieve Wikipedia passages that answer the question |
| T2Ranking | Given a Chinese search query, retrieve web passages that answer the question |
| TriviaQA | Retrieve Wikipedia passages that answer the question |

Table 8: Instructions used for each of the E5 datasets.

also holds true for other variants of the Mistral-7B model. We take this as additional evidence that the Mistral-7B base model was trained with some for of bidirectional attention.

# G Details on supervised results

## G.1 E5 dataset

The dataset consists of ELI5 (sample ratio 0.1) (Fan et al., 2019), HotpotQA (Yang et al., 2018), FEVER (Thorne et al., 2018), MIRACL (Zhang et al., 2023), MS-MARCO passage ranking (sample ratio 0.5) and document ranking (sample ratio 0.2) (), NQ (Karpukhin et al., 2020), NLI (Gao et al., 2021), SQuAD (Rajpurkar et al., 2016), TriviaQA (Joshi et al., 2017), Quora Duplicate Questions (sample ratio 0.1) (DataCanary et al., 2017), Mr- TyDi (Zhang et al., 2021), DuReader (He et al., 2018), and T2Ranking (sample ratio 0.5) (Xie et al., 2023). The instruction used for each dataset can be found in Table 8.

## G.2 Training details

All models are trained with LoRA $r = 16$ and $\alpha = 32$, brain floating point (bfloat16) quantization, gradient checkpointing, and flash attention 2 (Dao, 2024) to optimize GPU memory consumption. We train on 8 NVIDIA A100 GPUs with an effective batch size of 512 for 1000 steps using a maximum sequence length of 512 tokens. We use the Adam optimizer with a learning rate of $2e - 4$ and a linear learning rate warm-up for the first 300 steps.

| Rank | Model | Size (GB) | Public Data | Embed. Dim. | Retr. 15 | Rerank. 4 | Clust. 11 | PairClass. 3 | Class. 12 | STS 10 | Summ. 1 | Avg 56 |
|---|---|---|---|---|---|---|---|---|---|---|---|---|
| 1 | SFR-Embedding-Mistral | 14.22 | × | 4096 | 59.00 | 60.64 | 51.67 | 88.54 | 78.33 | 85.05 | 31.16 | 67.56 |
| 2 | voyage-lite-02-instruct | 2.45 | × | 1024 | 56.60 | 58.24 | 52.42 | 86.87 | 79.25 | 85.79 | 31.01 | 67.13 |
| 3 | GritLM-7B | 14.48 | × | 4096 | 57.41 | 60.49 | 50.61 | 87.16 | 79.46 | 83.35 | 30.37 | 66.76 |
| 4 | e5-mistral-7b-instruct | 14.22 | × | 4096 | 56.89 | 60.21 | 50.26 | 88.34 | 78.47 | 84.63 | 31.40 | 66.63 |
| 5 | GritLM-8x7B | 93.41 | × | 4096 | 55.09 | 59.80 | 50.14 | 84.97 | 78.53 | 83.26 | 29.82 | 65.66 |
| 6 | Bi + MNTP + Mean | 14.22 | ✓ | 4096 | 55.99 | 58.42 | 45.54 | 87.99 | 76.63 | 84.09 | 29.96 | 64.80 |
| 6 | Bi + MNTP + SimCSE + Mean | 14.22 | ✓ | 4096 | 56.05 | 58.59 | 45.12 | 88.18 | 76.72 | 83.69 | 30.66 | 64.72 |
| 7 | echo-mistral-7b-instruct-lasttoken | 14.22 | ✓ | 4096 | 55.52 | 58.14 | 46.32 | 87.34 | 77.43 | 82.56 | 30.73 | 64.68 |
| 8 | mxbai-embed-large-v1 | 0.67 | × | 1024 | 54.39 | 60.11 | 46.71 | 87.20 | 75.64 | 85.00 | 32.71 | 64.68 |
| 9 | UAE-Large-V1 | 1.34 | × | 1024 | 54.66 | 59.88 | 46.73 | 87.25 | 75.58 | 84.54 | 32.03 | 64.64 |
| 10 | text-embedding-3-large | - | × | 3072 | 55.44 | 59.16 | 49.01 | 85.72 | 75.45 | 81.73 | 29.92 | 64.59 |

Table 9: Top-10 models on the MTEB leaderboard as of 2024-03-29. LLM2Vec achieves the 6th rank overall, and the top rank among models trained with only publicly available data.

### G.3 Results

Table 2 presents the performance of applying `Bi + MNTP` and `Bi + MNTP + SimCSE` with mean pooling on MTEB benchmark. We also compare the performance of our models with recent and popular models trained with only publicly available data. We further report the current top-10 models in the MTEB leaderboard, including LLM2Vec$_{\texttt{Mistral-7B}}$ in Table 9. Our models achieve the 6th score in the MTEB leaderboard and the 1st among the models trained with only public data.

We present the detailed performance of supervised LLM2Vec-transformed models on full MTEB in Table 12. Here, we only report the `Bi + MNTP` transformed models as we showed they perform the best after supervised fine-tuning.

| Task Name | Instruction |
|---|---|
| AmazonCounterfactualClassif. | Classify a given Amazon customer review text as either counterfactual or not-counterfactual |
| AmazonPolarityClassification | Classify Amazon reviews into positive or negative sentiment |
| AmazonReviewsClassification | Classify the given Amazon review into its appropriate rating category |
| Banking77Classification | Given a online banking query, find the corresponding intents |
| EmotionClassification | Classify the emotion expressed in the given Twitter message into one of the six emotions: anger, fear, joy, love, sadness, and surprise |
| ImdbClassification | Classify the sentiment expressed in the given movie review text from the IMDB dataset |
| MassiveIntentClassification | Given a user utterance as query, find the user intents |
| MassiveScenarioClassification | Given a user utterance as query, find the user scenarios |
| MTOPDomainClassification | Classify the intent domain of the given utterance in task-oriented conversation |
| MTOPIntentClassification | Classify the intent of the given utterance in task-oriented conversation |
| ToxicConversationsClassif. | Classify the given comments as either toxic or not toxic |
| TweetSentimentClassification | Classify the sentiment of a given tweet as either positive, negative, or neutral |
| ArxivClusteringP2P | Identify the main and secondary category of Arxiv papers based on the titles and abstracts |
| ArxivClusteringS2S | Identify the main and secondary category of Arxiv papers based on the titles |
| BiorxivClusteringP2P | Identify the main category of Biorxiv papers based on the titles and abstracts |
| BiorxivClusteringS2S | Identify the main category of Biorxiv papers based on the titles |
| MedrxivClusteringP2P | Identify the main category of Medrxiv papers based on the titles and abstracts |
| MedrxivClusteringS2S | Identify the main category of Medrxiv papers based on the titles |
| RedditClustering | Identify the topic or theme of Reddit posts based on the titles |
| RedditClusteringP2P | Identify the topic or theme of Reddit posts based on the titles and posts |
| StackExchangeClustering | Identify the topic or theme of StackExchange posts based on the titles |
| StackExchangeClusteringP2P | Identify the topic or theme of StackExchange posts based on the given paragraphs |
| TwentyNewsgroupsClustering | Identify the topic or theme of the given news articles |
| SprintDuplicateQuestions | Retrieve duplicate questions from Sprint forum |
| TwitterSemEval2015 | Retrieve tweets that are semantically similar to the given tweet |
| TwitterURLCorpus | Retrieve tweets that are semantically similar to the given tweet |
| AskUbuntuDupQuestions | Retrieve duplicate questions from AskUbuntu forum |
| MindSmallReranking | Retrieve relevant news articles based on user browsing history |
| SciDocsRR | Given a title of a scientific paper, retrieve the titles of other relevant papers |
| StackOverflowDupQuestions | Retrieve duplicate questions from StackOverflow forum |
| ArguAna | Given a claim, find documents that refute the claim |
| ClimateFEVER | Given a claim about climate change, retrieve documents that support or refute the claim |
| CQADupstackRetrieval | Given a question, retrieve detailed question descriptions from Stackexchange that are duplicates to the given question |
| DBPedia | Given a query, retrieve relevant entity descriptions from DBPedia |
| FEVER | Given a claim, retrieve documents that support or refute the claim |
| FiQA2018 | Given a financial question, retrieve user replies that best answer the question |
| HotpotQA | Given a multi-hop question, retrieve documents that can help answer the question |
| MSMARCO | Given a web search query, retrieve relevant passages that answer the query |
| NFCorpus | Given a question, retrieve relevant documents that best answer the question |
| NQ | Given a question, retrieve Wikipedia passages that answer the question |
| QuoraRetrieval | Given a question, retrieve questions that are semantically equivalent to the given question |
| SCIDOCS | Given a scientific paper title, retrieve paper abstracts that are cited by the given paper |
| SciFact | Given a scientific claim, retrieve documents that support or refute the claim |
| Touche2020 | Given a question, retrieve detailed and persuasive arguments that answer the question |
| TRECCOVID | Given a query on COVID-19, retrieve documents that answer the query |
| STS* | Retrieve semantically similar text. |
| BUCC/Tatoeba | Retrieve parallel sentences. |
| SummEval | Given a news summary, retrieve other semantically similar summaries |

Table 10: Instructions used for evaluation on the MTEB benchmark. "STS*" refers to all the STS tasks.

| Task | S-LLaMA-1.3B | LLaMA-2-7B | Mistral-7B |
|------|--------------|------------|------------|
| AmazonCounterfactualClassification | 72.93 | 76.91 | 76.94 |
| AmazonPolarityClassification | 74.28 | 79.05 | 85.29 |
| AmazonReviewsClassification | 36.14 | 40.08 | 47.09 |
| ArguAna | 43.64 | 47.09 | 51.00 |
| ArxivClusteringP2P | 42.91 | 47.81 | 47.56 |
| ArxivClusteringS2S | 35.20 | 40.53 | 39.92 |
| AskUbuntuDupQuestions | 52.70 | 55.56 | 58.60 |
| BIOSSES | 75.12 | 82.41 | 83.29 |
| Banking77Classification | 79.00 | 84.65 | 86.16 |
| BiorxivClusteringP2P | 35.02 | 38.12 | 36.14 |
| BiorxivClusteringS2S | 27.21 | 31.25 | 30.26 |
| CQADupstackRetrieval | 18.50 | 30.78 | 33.37 |
| ClimateFEVER | 18.95 | 20.67 | 22.97 |
| DBPedia | 13.21 | 25.81 | 25.48 |
| EmotionClassification | 42.85 | 46.58 | 48.88 |
| FEVER | 16.96 | 43.48 | 45.11 |
| FiQA2018 | 16.99 | 24.62 | 27.24 |
| HotpotQA | 22.64 | 48.46 | 54.54 |
| ImdbClassification | 71.92 | 75.68 | 77.95 |
| MSMARCO | 7.03 | 18.81 | 19.13 |
| MTOPDomainClassification | 91.24 | 94.33 | 95.48 |
| MTOPIntentClassification | 74.08 | 79.54 | 82.84 |
| MassiveIntentClassification | 69.99 | 73.84 | 76.65 |
| MassiveScenarioClassification | 75.15 | 79.17 | 79.99 |
| MedrxivClusteringP2P | 30.15 | 30.94 | 30.11 |
| MedrxivClusteringS2S | 26.96 | 28.04 | 26.93 |
| MindSmallReranking | 29.52 | 30.86 | 29.73 |
| NFCorpus | 15.73 | 26.81 | 27.16 |
| NQ | 17.96 | 33.21 | 34.16 |
| QuoraRetrieval | 78.23 | 86.15 | 84.40 |
| RedditClustering | 38.67 | 42.84 | 41.83 |
| RedditClusteringP2P | 53.42 | 60.10 | 62.08 |
| SCIDOCS | 5.53 | 10.00 | 15.35 |
| SICK-R | 69.34 | 71.77 | 75.55 |
| STS12 | 60.09 | 65.39 | 67.65 |
| STS13 | 72.52 | 79.26 | 83.90 |
| STS14 | 66.70 | 72.98 | 76.97 |
| STS15 | 77.69 | 82.72 | 83.80 |
| STS16 | 75.94 | 81.02 | 81.91 |
| STS17 | 81.67 | 86.70 | 85.58 |
| STS22 | 63.70 | 63.47 | 65.93 |
| STSBenchmark | 73.36 | 78.32 | 80.42 |
| SciDocsRR | 67.76 | 77.62 | 77.81 |
| SciFact | 38.31 | 64.48 | 68.67 |
| SprintDuplicateQuestions | 77.36 | 87.57 | 91.30 |
| StackExchangeClustering | 59.35 | 65.12 | 67.34 |
| StackExchangeClusteringP2P | 31.47 | 33.61 | 34.50 |
| StackOverflowDupQuestions | 40.82 | 47.77 | 49.80 |
| SummEval | 31.23 | 31.38 | 30.19 |
| TRECCOVID | 56.04 | 60.67 | 55.66 |
| Touche2020 | 19.17 | 10.18 | 6.54 |
| ToxicConversationsClassification | 68.41 | 71.81 | 70.71 |
| TweetSentimentExtractionClassification | 56.08 | 57.17 | 60.90 |
| TwentyNewsgroupsClustering | 31.54 | 30.76 | 30.26 |
| TwitterSemEval2015 | 61.54 | 65.14 | 68.76 |
| TwitterURLCorpus | 77.73 | 80.94 | 82.76 |
| Average | 49.42 | 55.36 | 56.80 |

Table 11: Unsupervised results of LLM2Vec transformed models on MTEB.

| Task | S-LLaMA-1.3B | LLaMA-2-7B | Mistral-7B |
|---|---|---|---|
| AmazonCounterfactualClassification | 77.42 | 82.22 | 77.58 |
| AmazonPolarityClassification | 82.05 | 89.69 | 91.12 |
| AmazonReviewsClassification | 40.81 | 48.47 | 49.97 |
| ArguAna | 51.66 | 56.53 | 57.48 |
| ArxivClusteringP2P | 43.47 | 43.14 | 42.81 |
| ArxivClusteringS2S | 39.85 | 42.38 | 44.24 |
| AskUbuntuDupQuestions | 60.71 | 63.13 | 63.98 |
| BIOSSES | 85.88 | 82.13 | 85.24 |
| Banking77Classification | 86.01 | 88.17 | 88.31 |
| BiorxivClusteringP2P | 37.10 | 35.88 | 34.27 |
| BiorxivClusteringS2S | 34.28 | 34.81 | 35.53 |
| CQADupstackRetrieval | 41.73 | 45.94 | 48.84 |
| ClimateFEVER | 33.49 | 30.70 | 35.19 |
| DBPedia | 43.58 | 48.42 | 49.58 |
| EmotionClassification | 48.38 | 51.71 | 52.05 |
| FEVER | 86.81 | 89.93 | 89.40 |
| FiQA2018 | 41.00 | 51.28 | 53.11 |
| HotpotQA | 63.85 | 72.99 | 74.07 |
| ImdbClassification | 75.33 | 85.78 | 87.42 |
| MSMARCO | 38.32 | 41.45 | 42.17 |
| MTOPDomainClassification | 94.09 | 95.57 | 96.04 |
| MTOPIntentClassification | 77.05 | 82.81 | 84.77 |
| MassiveIntentClassification | 75.58 | 78.06 | 79.29 |
| MassiveScenarioClassification | 79.16 | 81.35 | 81.64 |
| MedrxivClusteringP2P | 33.55 | 32.23 | 31.07 |
| MedrxivClusteringS2S | 31.11 | 31.37 | 31.27 |
| MindSmallReranking | 31.96 | 31.34 | 31.50 |
| NFCorpus | 37.12 | 40.33 | 39.33 |
| NQ | 53.89 | 61.24 | 61.70 |
| QuoraRetrieval | 87.37 | 85.59 | 87.75 |
| RedditClustering | 53.02 | 61.10 | 60.24 |
| RedditClusteringP2P | 60.47 | 64.52 | 64.12 |
| SCIDOCS | 17.96 | 21.05 | 22.50 |
| SICK-R | 82.25 | 83.01 | 83.70 |
| STS12 | 78.28 | 78.85 | 78.80 |
| STS13 | 85.52 | 86.84 | 86.37 |
| STS14 | 82.49 | 84.04 | 84.04 |
| STS15 | 88.76 | 88.72 | 88.99 |
| STS16 | 87.11 | 86.79 | 87.22 |
| STS17 | 90.10 | 90.63 | 90.19 |
| STS22 | 68.25 | 67.55 | 67.68 |
| STSBenchmark | 87.16 | 88.72 | 88.65 |
| SciDocsRR | 79.23 | 84.03 | 83.80 |
| SciFact | 72.08 | 77.30 | 78.86 |
| SprintDuplicateQuestions | 96.25 | 96.83 | 96.82 |
| StackExchangeClustering | 63.04 | 67.98 | 70.73 |
| StackExchangeClusteringP2P | 34.01 | 33.20 | 34.50 |
| StackOverflowDupQuestions | 49.61 | 51.02 | 54.41 |
| SummEval | 30.01 | 28.49 | 29.96 |
| TRECCOVID | 80.41 | 79.25 | 77.69 |
| Touche2020 | 22.31 | 16.92 | 22.18 |
| ToxicConversationsClassification | 69.92 | 71.01 | 69.26 |
| TweetSentimentExtractionClassification | 60.76 | 61.11 | 62.14 |
| TwentyNewsgroupsClustering | 49.37 | 51.04 | 52.18 |
| TwitterSemEval2015 | 76.14 | 80.70 | 80.60 |
| TwitterURLCorpus | 86.23 | 86.56 | 86.56 |
| Average | 61.85 | 64.14 | 64.80 |

Table 12: Supervised results of LLM2Vec (only `Bi` + `MNTP`) models on MTEB.

