# OpenReview forum: "LLM2Vec: Large Language Models Are Secretly Powerful Text Encoders"
_colmweb.org/COLM/2024/Conference — COLM_

### Official Review · Reviewer_cJbi · 2024-05-09

**Rating:** 6
**Confidence:** 5
**Ethics Flag:** 1

**Summary:**

This paper studies how to adapt LLMs to embedding models. The proposed method converts an pre-trained auto-regressive language models by changing causal attention into bidirectional attention, and then finetune the model with masked next token prediction and unsupervised contrastive learning. Experimental results show that the proposed method achieves strong performance under both unsupervised and supervised settings.

**Reasons To Accept:**

1. The paper presents a easy-to-follow recipe to get strong embedding models from pre-trained LLMs.
2. The empirical results are strong.

**Reasons To Reject:**

1. It can hardly be considered a secret that LLM can be strong embedding models, as previous works like E5-mistral-instruct have already demonstrated similar findings.
2. Under supervised setting, the proposed bidirectional attention and unsupervised training provides little improvements over the replicated E5-mistral-instruct model.

---

> ### Author Rebuttal · Authors · 2024-05-29
>
> We thank the reviewer for their efforts and for pointing out that our paper provides an easy-to-follow method which leads to strong empirical results. We address the concerns raised by the reviewer below.
>
> **Not a secret**: The reviewer is correct in pointing out that the E5-Mistral model trained by Wang et al. (2024) is also based on a decoder-only LLM. However, there are a few important points worth highlighting. First, part of our motivation for saying “secretly” in the title is the strong performance of our method in the unsupervised setting and the fact that only little adaptation is needed to convert decoder-only LLMs into strong embedders. This hasn’t been extensively explored yet and the simplicity of our approach shows that you don’t necessarily need supervised contrastive fine-tuning to convert LLMs into strong embedding models. Second, the paper by Wang et al. 2024 is in fact concurrent work to ours (it was released less than 3 months before the COLM deadline). That said, we are open to suggestions for how to update our title.
>
> **Regarding the improvements in the supervised setting**: We highlight that the MTEB scores are average scores across 56 tasks. Therefore, even a small improvement corresponds to improvement over various tasks in different categories, as shown in category-detailed results.  In Table 1 and 2. Stated differently, this is a property of the MTEB benchmark. To provide evidence for this, we computed the performance improvement for each of the top-10 models on the MTEB leaderboard provided in Table 9 in the Appendix. It can be seen that each entry reports a “small” improvement over the previous approach and LLM2Vec also follows this pattern: 10) text-embedding-3-large (64.59) – +0.077% – 9) UAE-Large-V1 (64.64) – +0.062% – 8) mxbai-embed-large-v1 (64.68) – + 0.0% – 7) echo-mistral-7b-instruct-lasttoken (64.68) – +0.186% – 6) LLM2Vec (Bi+MNTP+Mean) (64.80) – +1.327% – 5) GritLM-8x7B (65.66) – +1.477% – 4) e5-mistral-7b-instruct (66.63) – +0.195% – 3) GritLM-7B (66.76) – +0.554% – 2) voyage-lite-02-instruct (67.13) – +0.641% – 1) SFR-Embedding-Mistral (67.56). Note that the improvements from rank 5) onwards are expected to be bigger compared to models on ranks 10) to 6) as these models rely on synthetic data for supervised fine-tuning.
>
> We hope that we could clarify the issues raised by the reviewer and remain available for further clarifications.
>
> Wang et al. (2024) - Improving Text Embeddings with Large Language Models

---

### Official Review · Reviewer_ihtZ · 2024-05-10

**Rating:** 7
**Confidence:** 4
**Ethics Flag:** 1

**Summary:**

The article presents LLM2Vec, a novel technique to transform large, decoder-only language models into efficient text encoders without the need for expensive adaptation or the creation of synthetic GPT-4 data.
This approach consists of three simple steps: enabling bidirectional attention, masked next token prediction (MNTP), and unsupervised contrastive learning.
The authors evaluate the effectiveness of LLM2Vec on various English-language word- and sequence-level tasks.

**Questions To Authors:**

It would be beneficial if future work could address the transferability of the LLM2Vec method to models trained in other languages. This would strengthen the generalizability of the approach.

**Reasons To Accept:**

**Innovation**: LLM2Vec introduces a new methodology to leverage pre-trained decoder-only language models, paving the way for better text embedding performance.    In particular, MNTP task, well adapted to the encoder and decoder.

 **Simplicity**: The approach is straightforward with only three steps, making it easy to replicate and understand.

**Performance**: The method demonstrates superior performance in unsupervised settings, as well as in comparisons with supervised contrastive learning, establishing new state-of-the-art results.

**Reasons To Reject:**

**Language Limitation**: The evaluation is performed solely on English datasets, which does not guarantee that the method will generalize across other languages.

---

> ### Author Rebuttal · Authors · 2024-05-29
>
> We thank the reviewer for their positive assessment of our work. We are happy to see that the reviewer points out the simplicity of our proposed approach as well as its strong performance.
>
> **Regarding the language limitation**: We want to point out that we are aware of this limitation and also discuss it in the limitation section of our paper (see Appendix A on page 16). We believe that our paper, as is, makes a strong contribution to LLM-based text representation and given the simplicity of our approach, we are hopeful that it can be extended to more languages. In fact, we are already working on this.

---

> > ### Comment · Reviewer_ihtZ · 2024-06-05
> >
> > I have read the response as well as review comments from others and will keep my score unchanged.

---

### Official Review · Reviewer_1qid · 2024-05-12

**Rating:** 5
**Confidence:** 4
**Ethics Flag:** 1

**Summary:**

This paper introduces an unsupervised approach to convert decoder-only LLMs into robust text encoders. This approach, named LLM2Vec, is structured around three main steps: enabling bidirectional attention, masked next token prediction (MNTP), and unsupervised contrastive learning. The authors claim that LLM2Vec outperforms existing encoder-only models on various NLP tasks and establishes new benchmarks on the Massive Text Embeddings Benchmark (MTEB) using publicly available data.

**Questions To Authors:**

How does LLM2Vec impact the interpretability of the embeddings? Given that the model transformations introduce changes in how tokens are encoded, does this affect the ability to trace back and understand decisions made by the model?

**Reasons To Accept:**

1. The authors provide extensive empirical evidence demonstrating that LLM2Vec enhances the performance of LLMs on both word-level and sequence-level embedding tasks, surpassing existing state-of-the-art models.

2. LLM2Vec is described as data- and parameter-efficient, which is particularly important for practical applications where computational resources may be limited.

**Reasons To Reject:**

1. **Innovation and Originality**: While the paper skillfully integrates several established techniques—bidirectional attention, masked next token prediction (MNTP), and unsupervised contrastive learning—into the LLM2Vec framework, it does not introduce fundamentally new methodologies. The core contribution lies in the novel application and combination of these existing methods rather than in the development of new techniques. This approach might raise questions about the paper's novelty within the field, particularly in a research landscape that highly values groundbreaking innovations. A more detailed discussion on the unique configurations or optimizations made to these existing techniques within LLM2Vec could enhance the perceived novelty of the work.

2. **Fairness in Comparative Analysis**: The paper implements a transformation from unidirectional to bidirectional attention in decoder-only LLMs to enhance text embedding capabilities. However, the comparative analysis appears limited as the baseline configurations in the unidirectional setting (labeled as Uni+Mean) do not include enhancements comparable to those applied in the bidirectional setting (Bi+MNTP+SimCSE+Mean). A more equitable comparison would involve aligning the enhancements across both settings, for instance, comparing Uni+MNTP+SimCSE+Mean with Bi+MNTP+SimCSE+Mean. This would provide a clearer assessment of the impact of changing the attention mechanism independently of other variables.

---

> ### Author Rebuttal · Authors · 2024-05-29
>
> We thank the reviewer for their feedback and are happy to see that they found that our paper provides extensive empirical results with data- and parameter-efficient methods which is particularly relevant for practical aspects. We address the reviewers' concerns below.
>
> **Innovation and originality**: We believe that showing how to combine already-established methods in a novel setting, i.e., the application of LLMs for general text representations, is a novel contribution. We will follow the reviewers’ suggestion and will update the final version with more discussion on the adaptations we made to make these methods work in our setting. For instance, we substituted the masked token prediction of MLM with masked next token prediction due to the nature of auto-regressive language models, i.e., predicting the next instead of the current token.
>
> **Fairness in comparative analysis**: MNTP is specifically designed for bidirectional models and to adapt them to make use of future tokens, which is why we do not use it in the unidirectional case. While in theory MNTP could be applied to unidirectional models, it would still suffer from the limitation of only having access to the past, and therefore we would not expect it to provide any benefit over the original LM objective. That said, we already provide additional configurations (Uni and Uni+SimCSE for three different pooling methods) for unidirectional approaches in Table 5 in the Appendix, where we also ablate the importance of each component of LLM2Vec recipe.
>
> **Regarding the question about interpretability**: Existing LM interpretability methods for input attribution or for analyzing attention patterns could be applied on top of LLM2Vec transformed models. However, this is beyond the scope of our paper and we leave such investigations for future work. That said, section 4 of our paper already provides an extensive analysis of how bidirectionality and the LLM2Vec recipe affect the representations of LLMs.
>
> We hope that our response is sufficient to address the concerns raised by the reviewer and that they consider revisiting their score in light of our clarifications. We remain available for discussions.

---

> > ### Comment · Reviewer_1qid · 2024-06-06
> >
> > I have read the response and review comments from others and updated my score.

---

### Official Review · Reviewer_pAEP · 2024-05-13

**Rating:** 6
**Confidence:** 3
**Ethics Flag:** 1

**Summary:**

This paper introduces a method to transform any decoder-only LLM into a bidirectional encoder.  The paper evaluates the method on three different LLMs and results show the effectiveness of the method.

**Reasons To Accept:**

The method is simple to understand and the writing is clear. The results of the experiments are significant. The analysis is  also clear.

**Reasons To Reject:**

However, there are some disadvantages in this paper.

1. Authors claim that the encoder building by decoder-only LLM can generate rich contextualized representations for NLP tasks. Although experiments compare the encoder with LLMs, LLMs may perform better with other prompts. It is better to compare models with other prompts.

2. The method is described clearly in this paper while the method is similar to MLM from BERT and other models. Authors should give some ablations to show the effectiveness of MNTP.

3. As discussed above, LLMs can label input sequences by generating sequences of labels. Author should make it clear that LLMs labels input sequences by generating or directly labeling tokens.

---

> ### Author Rebuttal · Authors · 2024-05-29
>
> We want to thank the reviewer for their positive feedback as they find our paper simple to understand, providing an effective method, significant experiments, and clear analysis. We address the concerns raised by the reviewer below.
>
> **Using other prompts**: Springer et al. (2024) have shown that LLMs for text embeddings are robust to minor prompt variations. To ensure a fair comparison between our method and the baselines, we did perform our experiments using the best performing instructions which are also used in previous work (Echo-embeddings, E5, and GritLM) for all models.
>
> Additionally, we conducted an experiment to quantify the change in performance resulting from prompt variations. We compare the evaluation results for two sets of instructions – the ones we use in our paper (referred to as E5 inst) and those used in Su et al. (2022) (referred to as Instructor inst). In general, E5 inst are more detailed than Instructor inst. We ran the evaluation on the subset of 15 MTEB tasks described in Appendix C.1. We compare our best unsupervised approach without any contrastive learning (Bi + MNTP + Mean)  with Echo.
>
> We observe that for both approaches, the performance drops by 2-3% when switching from E5 inst to Instructor inst. So as expected both approaches exhibit highly similar behavior when changing prompts (this is expected as they use the same underlying LLM). Finally, we want to emphasize again that the prompts we are using in the paper (E5 inst) are the currently best performing prompts in the literature. We will make sure to include these results in the final version.
>
> **Ablations for MNTP**: We would like to point out that the ablation the reviewer is asking for is already provided in the paper. Table 5 in the appendix shows the improvements gained by each step, including MNTP. To be more precise, the impact of MNTP can be observed by comparing Bi vs. Bi+MNTP and Bi+SimCSE vs. Bi+MNTP+SimCSE models when mean pooling is applied. For instance, applying MNTP before SimCSE improves Llama-2’s performance by 34% in the unsupervised setting.
>
> We kindly ask the reviewer to elaborate more on their third point so that we can address their concern.
>
> We hope that the reviewer considers revising their score based on our answer and we remain available for further clarifications during the discussion period.
>
> Su et al. (2022) - One Embedder, Any Task: Instruction-Finetuned Text Embeddings
> Springer et al. (2024) - Repetition Improves Language Model Embeddings

---

### Decision · Program_Chairs · 2024-07-10

**Decision:**

Accept

**Comment:**

This paper introduces LLM2Vec, an unsupervised method to transform decoder-only LLMs into powerful text encoders. LLM2Vec employs three core steps: enabling bidirectional attention, predicting masked next tokens, and applying contrastive learning without labels. The authors demonstrate this method's effectiveness on various word-level and sequence-level tasks.

Pros:
- The method is easy to understand, and the paper is well-structured and readable.
- The authors provide substantial empirical evidence showcasing the model's superior performance.

Cons:
- Reviewers primarily criticized the paper's novelty. It combines well-known techniques—bidirectional attention, masked next token prediction (MNTP), and unsupervised contrastive learning.

I personally believe that integrating established techniques in a new framework is an excellent approach to creating a simple yet effective method.